# Bypassing Feature Squeezing by Increasing Adversary Strength

**Yash Sharma[1] and Pin-Yu Chen[2]**

[1]The Cooper Union, New York, NY 10003, USA
[2]IBM Research, Yorktown Heights, NY 10598, USA
sharma2@cooper.edu, pin-yu.chen@ibm.com

## Abstract

Feature Squeezing is a recently proposed defense method which reduces the search space available to an adversary by coalescing samples that correspond to many different feature vectors in the original space into a single sample. It has been shown that feature squeezing defenses can be combined in a joint detection framework to achieve high detection rates against state-of-the-art attacks. However, we demonstrate on the MNIST and CIFAR-10 datasets that by increasing the adversary strength of said state-of-the-art attacks, one can bypass the detection framework with adversarial examples of minimal visual distortion. These results suggest for proposed defenses to validate against stronger attack configurations.

## 1 Introduction

Deep neural networks (DNNs) achieve state-of-the-art performance in various tasks in machine learning and artificial intelligence, such as image classification, speech recognition, machine translation and game-playing. Despite their effectiveness, recent studies have illustrated the vulnerability of DNNs to adversarial examples Szegedy et al. (2013); Goodfellow et al. (2015). For instance, a carefully designed perturbation to an image can lead a well-trained DNN to misclassify. Even worse, effective adversarial examples can also be made virtually indistinguishable to human perception. Adversarial examples crafted to evade a specific model can even be used to mislead other models trained for the same task, exhibiting a property known as transferability Liu et al. (2016); Papernot et al. (2016); Sharma & Chen (2017).

To address this problem, numerous defense mechanisms have been proposed; one which has achieved strong results is feature squeezing. Feature squeezing relies on applying input transformations to reduce the degrees of freedom available to an adversary by "squeezing" out unnecessary input features. The authors in Xu et al. (2017) propose a detection method using such input transformations by relying on the intuition that if the original and squeezed inputs produce substantially different outputs from the model, the input is likely to be adversarial. By comparing the difference between predictions with a selected threshold value, the system is designed to output the correct prediction for legitimate examples and reject adversarial inputs. By combining multiple squeezers in a joint detection framework, the authors claim that the system can can successfully detect adversarial examples from eleven state-of-the-art methods Xu et al. (2017).

In this paper, we show that by increasing the adversary strength of the state-of-the-art methods, the feature squeezing joint detection method can be readily bypassed. We demonstrate this on both the MNIST and CIFAR-10 datasets. We experiment with EAD Chen et al. (2017), a generalization of the state-of-the-art C&W $L_2$ attack Carlini & Wagner (2017), and PGD Madry et al. (2017), an $L_\infty$ attack motivated to be the strongest adversary using the local first-order information about the network. For EAD, and C&W, we increase the adversary strength by increasing $\kappa$, which controls the necessary margin between the predicted probability of the target class and that of the rest. For PGD, we increase the adversary strength by increasing $\epsilon$, which controls the allowable $L_\infty$ distortion. We find that adversarial examples with minimal visual distortion can be generated which bypass feature squeezing under these stronger attack configurations. Our results suggest that proposed defenses

Table 1: Comparison of PGD, C&W, and EAD results against the MNIST joint detector at various confidence levels. ASR means attack success rate (%). The distortion metrics are averaged over successful examples.

| | | Non-Targeted | | | | Targeted | | | | | | |
| | | | | | | Next | | | | LL | | |
| Attack Method | Confidence | ASR | $L_1$ | $L_2$ | $L_\infty$ | ASR | $L_1$ | $L_2$ | $L_\infty$ | ASR | $L_1$ | $L_2$ | $L_\infty$ |
|---|---|---|---|---|---|---|---|---|---|---|---|---|---|
| PGD | None | 100% | 196.0 | 10.17 | 0.900 | 78% | 169.8 | 8.225 | 0.881 | 67% | 188.1 | 9.091 | 0.991 |
| C&W | 10 | 0% | 21.05 | 1.962 | 0.568 | 0% | 31.94 | 2.748 | 0.655 | 0% | 37.78 | 3.207 | 0.732 |
| | 20 | 15% | 27,21 | 2.472 | 0.665 | 10% | 40.51 | 3.419 | 0.763 | 24% | 47.86 | 3.977 | 0.820 |
| | 30 | 64% | 34.30 | 3.019 | 0.754 | 67% | 47.43 | 3.973 | 0.842 | 91% | 59.56 | 4.811 | 0.888 |
| | 40 | 87% | 42.04 | 3.590 | 0.831 | 97% | 61.12 | 4.938 | 0.922 | 100% | 72.88 | 5.715 | 0.939 |
| EAD | 10 | 24% | 11.44 | 2.286 | 0.879 | 7% | 19.69 | 3.114 | 0.942 | 7% | 23.99 | 3.481 | 0.955 |
| | 20 | 80% | 15.26 | 2.766 | 0.921 | 65% | 26.80 | 3.752 | 0.964 | 78% | 31.81 | 4.122 | 0.972 |
| | 30 | 95% | 20.17 | 3.264 | 0.957 | 97% | 35.50 | 4.449 | 0.983 | 93% | 39.68 | 4.769 | 0.991 |
| | 40 | 97% | 26.50 | 3.803 | 0.972 | 100% | 44.75 | 5.114 | 0.992 | 100% | 50.21 | 5.532 | 0.997 |

Table 2: Comparison of PGD, C&W, and EAD results against the CIFAR-10 joint detector at various confidence levels. ASR means attack success rate (%). The distortion metrics are averaged over successful examples.

| | | Non-Targeted | | | | Targeted | | | | | | |
| | | | | | | Next | | | | LL | | |
| Attack Method | Confidence | ASR | $L_1$ | $L_2$ | $L_\infty$ | ASR | $L_1$ | $L_2$ | $L_\infty$ | ASR | $L_1$ | $L_2$ | $L_\infty$ |
|---|---|---|---|---|---|---|---|---|---|---|---|---|---|
| PGD | None | 100% | 81.18 | 1.833 | 0.070 | 100% | 212.0 | 4.979 | 0.299 | 100% | 214.9 | 5.042 | 0.300 |
| C&W | 10 | 32% | 10.51 | 0.274 | 0.033 | 0% | 14.25 | 0.368 | 0.042 | 0% | 17.36 | 0.445 | 0.049 |
| | 30 | 78% | 28.80 | 0.712 | 0.073 | 51% | 37.11 | 0.901 | 0.083 | 6% | 41.51 | 1.006 | 0.093 |
| | 50 | 96% | 59.32 | 1.416 | 0.130 | 98% | 82.54 | 1.954 | 0.169 | 94% | 90.17 | 2.129 | 0.179 |
| | 70 | 100% | 120.2 | 2.827 | 0.243 | 100% | 201.2 | 4.713 | 0.375 | 100% | 212.2 | 4.962 | 0.403 |
| EAD | 10 | 46% | 6.371 | 0.379 | 0.079 | 10% | 8.187 | 0.508 | 0.109 | 0% | 10.17 | 0.597 | 0.121 |
| | 30 | 78% | 18.94 | 0.876 | 0.146 | 51% | 25.98 | 1.090 | 0.166 | 23% | 29.58 | 1.209 | 0.175 |
| | 50 | 94% | 42.36 | 1.550 | 0.206 | 96% | 62.90 | 2.094 | 0.247 | 90% | 70.23 | 2.296 | 0.275 |
| | 70 | 100% | 83.14 | 2.670 | 0.317 | 100% | 157.9 | 4.466 | 0.477 | 100% | 172.8 | 4.811 | 0.502 |

should validate against adversarial examples of maximal distortion, as long as the examples remain visually adversarial.

## 2 EXPERIMENT SETUP

Two types of feature squeezing were focused on by the authors in Xu et al. (2017): (i) reducing the color bit depth of images; and (ii) using smoothing (both local and non-local) to reduce the variation among pixels. For the detection method, the model's original prediction is compared with the prediction on the squeezed sample using the $L_1$ norm. As a defender typically does not know the exact attack method, multiple feature squeezers are combined by outputting the maximum distance. The threshold is selected targeting a false positive rate below 5% by choosing a threshold that is exceeded by no more than 5% of legitimate samples.

For MNIST, the joint detector consists of a 1-bit depth squeezer with 2x2 median smoothing. For CIFAR-10, the joint detector consists of a 5-bit depth squeezer with 2x2 median smoothing and a non-local means filter with a 13x13 search window, 3x3 patch size, and a Gaussian kernel bandwidth size of 2. We use the same thresholds as used in Xu et al. (2017). We generate adversarial examples using EAD Chen et al. (2017) and PGD attacks Madry et al. (2017).

EAD generalizes the state-of-the-art C&W $L_2$ attack Carlini & Wagner (2017) by performing elastic-net regularization, linearly combining the $L_1$ and $L_2$ penalty functions Chen et al. (2017). The hyperparameter $\beta$ controls the trade-off between $L_1$ and $L_2$ minimization. We test EAD in both the general case and the special case where $\beta$ is set to 0, which is equivalent to the C&W $L_2$ attack. For MNIST and CIFAR-10, $\beta$ was set to 0.01 and 0.001, respectively. We tune $\kappa$, which is a *confidence* parameter that controls the necessary margin between the predicted probability of the target class and that of the rest, in our experiments. $\kappa$ is increased starting from 10 on both datasets, which was the value used in the feature squeezing experiments Xu et al. (2017). Full detail on the implementation is provided in the supplementary material.

For $L_\infty$ attacks, which we will consider, fast gradient methods (FGM) use the sign of the gradient $\nabla J$ of the training loss $J$ with respect to the input for crafting adversarial examples Goodfellow

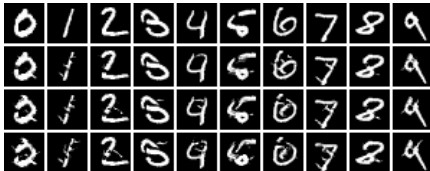



Figure 1: Randomly selected set of non-targeted MNIST adversarial examples generated by EAD. First row: Original, Subsequent rows: $\kappa = \{10, 20, 30\}$.

Figure 2: Randomly selected set of non-targeted CIFAR-10 adversarial examples generated by EAD. First row: Original, Subsequent rows: $\kappa = \{10, 30, 50\}$.

et al. (2015). PGD iteratively uses FGM with a finer distortion, followed by an $\epsilon$-ball clipping Madry et al. (2017). We tune $\epsilon$, which controls the allowable $L_\infty$ distortion, in our experiments. $\epsilon$ is increased starting from 0.3 on MNIST and 0.008 on CIFAR-10, which were the values used for the feature squeezing experiments Xu et al. (2017). Full detail on the implementation is provided in the supplementary material.

We randomly sample 100 images from the MNIST and CIFAR-10 test sets. For each dataset, we use the same pre-trained state-of-the-art models as used in Xu et al. (2017). We generate adversarial examples in the non-targeted case, force network to misclassify, and in the targeted case, force network to misclassify to a target class $t$. As done in Xu et al. (2017), we try two different targets, the *Next* class ($t = label + 1$ mod *# of classes*) and the least-likely class (*LL*).

## 3 EXPERIMENT RESULTS

The generated adversarial examples are tested against the proposed MNIST and CIFAR-10 joint detection configurations. In Tables 1 and 2, the results of tuning $\kappa$ for C&W and EAD are provided, and are presented with the results for PGD at the lowest $\epsilon$ value at which the highest attack success rate (ASR) was yielded, against the MNIST and CIFAR-10 joint detectors, respectively. In all cases, EAD outperforms the C&W $L_2$ attack, particularly at lower confidence levels, indicating the importance of minimizing the $L_1$ distortion for generating robust adversarial examples with minimal visual distortion. Specifically, we find that with enough strength, each attack is able to achieve near 100% ASR against the joint detectors.

In Figure 1, non-targeted MNIST adversarial examples generated by EAD are shown at $\kappa = \{10, 20, 30\}$. In Figure 2, non-targeted CIFAR-10 adversarial examples generated by EAD are shown at $\kappa = \{10, 30, 50\}$. Adversarial examples generated in the least-likely targeted case are provided in the supplementary material, These figures indicate that adversarial examples generated by EAD at high $\kappa$, which bypass the joint feature squeezing detector, have minimal visual distortion. This holds true for adversarial examples generated by PGD with high $\epsilon$ on CIFAR-10, but not on MNIST.

## 4 CONCLUSION

Feature Squeezing is a recently proposed class of input transformations which when combined in a joint detection framework has been shown to achieve high detection rates against state-of-the-art attacks. We show on the MNIST and CIFAR-10 datasets that by increasing the adversary strength, by tuning the confidence $\kappa$ and $L_\infty$ constraint $\epsilon$ for EAD and PGD, respectively, the proposed joint detection configuration can be bypassed with adversarial examples of minimal visual distortion. These results suggest for proposed defenses to validate against stronger attack configurations, using the maximal adversary strength where examples remain visually similar to the inputs. For future work, we aim to validate if other recently proposed defenses are robust to strong adversaries.

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

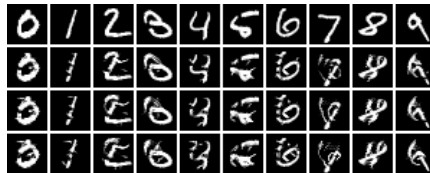
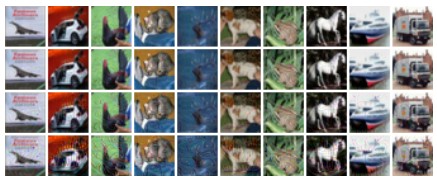

Figure 3: Randomly selected set of least-likely targeted MNIST adversarial examples generated by EAD. First row: Original, Subsequent rows: $\kappa = \{10, 20, 30\}$.

Figure 4: Randomly selected set of least-likely targeted CIFAR-10 adversarial examples generated by EAD. First row: Original, Subsequent rows: $\kappa = \{10, 30, 50\}$.

## 5 SUPPLEMENTARY MATERIAL

### 5.1 ATTACK DETAILS

The targeted attack formulations are discussed below, non-targeted attacks can be implemented in a similar fashion. We denote by $\mathbf{x}_0$ and $\mathbf{x}$ the original and adversarial examples, respectively, and denote by $t$ the target class to attack.

#### 5.1.1 EAD

EAD generalizes the state-of-the-art C&W $L_2$ attack Carlini & Wagner (2017) by performing elastic-net regularization, linearly combining the $L_1$ and $L_2$ penalty functions Chen et al. (2017). The formulation is as follows:

$$\text{minimize}_{\mathbf{x}} \quad c \cdot f(\mathbf{x}, t) + \beta \|\mathbf{x} - \mathbf{x}_0\|_1 + \|\mathbf{x} - \mathbf{x}_0\|_2^2$$
$$\text{subject to} \quad \mathbf{x} \in [0, 1]^p, \tag{1}$$

where $f(x, t)$ is defined as:

$$f(\mathbf{x}, t) = \max\{\max_{j \neq t}[\mathbf{Logit}(\mathbf{x})]_j - [\mathbf{Logit}(\mathbf{x})]_t, -\kappa\}, \tag{2}$$

By increasing $\beta$, one trades off $L_2$ minimization for $L_1$ minimization. When $\beta$ is set to 0, EAD is equivalent to the C&W $L_2$ attack. By increasing $\kappa$, one increases the necessary margin between the predicted probability of the target class and that of the rest. Therefore, increasing $\kappa$ improves adversary strength but compromises visual quality.

We implement 9 binary search steps on the regularization parameter $c$ (starting from 0.001) and run $I = 1000$ iterations for each step with the initial learning rate $\alpha_0 = 0.01$. For finding successful adversarial examples, we use the ADAM optimizer for the C&W attack and implement the projected FISTA algorithm with the square-root decaying learning rate for EAD Kingma & Ba (2014); Beck & Teboulle (2009).

#### 5.1.2 PGD

Fast gradient methods (FGM) use the gradient $\nabla J$ of the training loss $J$ with respect to $\mathbf{x}_0$ for crafting adversarial examples Goodfellow et al. (2015). For $L_\infty$ attacks, which we will consider, $\mathbf{x}$ is crafted by

$$\mathbf{x} = \mathbf{x}_0 - \epsilon \cdot \text{sign}(\nabla J(\mathbf{x}_0, t)), \tag{3}$$

where $\epsilon$ specifies the $L_\infty$ distortion between $\mathbf{x}$ and $\mathbf{x}_0$, and $\text{sign}(\nabla J)$ takes the sign of the gradient.

PGD iteratively uses FGM with a finer distortion, followed by an $\epsilon$-ball clipping Madry et al. (2017). 10 steps are used, and the step-size was set to be $\epsilon/10$, which has been shown to be an effective attack setting in Tramèr et al. (2017).

### 5.2 EAD ADVERSARIAL EXAMPLES IN THE TARGETED CASE (FIGURES 3 AND 4)

In Figure 3, least-likely targeted MNIST adversarial examples generated by EAD are shown at $\kappa = \{10, 20, 30\}$. In Figure 4, least-likely targeted CIFAR-10 adversarial examples generated by





Figure 5: Randomly selected set of non-targeted MNIST adversarial examples generated by PGD.
First row: Original, Subsequent rows: $\epsilon = \{0.3, 0.4, 0.5, 0.6, 0.7, 0.8.0.9, 1.0\}$.

Figure 6: Randomly selected set of non-targeted CIFAR-10 adversarial examples generated by PGD.
First row: Original, Subsequent rows: $\epsilon = \{0.008, 0.04, 0.07, 0.1, 0.2, 0.3, 0.4, 0.5\}$.

EAD are shown at $\kappa = \{10, 30, 50\}$. Distortion is more apparent in the targeted case, particularly in the least-likely targeted case, but the examples are still visually adversarial.

## 5.3 PGD ADVERSARIAL EXAMPLES IN THE NON-TARGETED CASE (FIGURES 5 AND 6)

In Figure 5, non-targeted MNIST adversarial examples generated by PGD are shown at $\epsilon = \{0.3, 0.4, 0.5, 0.6, 0.7, 0.8.0.9, 1.0\}$. In Figure 6, non-targeted CIFAR-10 adversarial examples generated by PGD are shown at $\epsilon = \{0.008, 0.04, 0.07, 0.1, 0.2, 0.3, 0.4, 0.5\}$. CIFAR-10 adversarial examples at high $\epsilon$ have minimal visual distortion, however MNIST examples at high $\epsilon$, which yield the optimal ASR, have clear distortion.

