# OpenReview forum: "Bypassing Feature Squeezing by Increasing Adversary Strength"
_ICLR.cc/2018/Workshop — Reject_

### Official Review · AnonReviewer3 · 2018-03-08
**This paper provides experiments that show the limits of feature squeezing defense**

**Rating:** 6
**Confidence:** 3

**Review:**

By increasing the strength of adversarial attacks by tuning their parameters out of the standard range, the provided experiments on MIST and CIFAR-10 show that feature squeezing defense can be by passed.

The issue of this claim is that if there are no limit range for the adversarial attacks parameters, it is obviously possible to change the input image in a way that the image class changes.
The author uses an implicit limit range that corresponds to a soft criteria corresponding to visual similarity.
Could you quantify "visual similarity" ?
What means L_2, L_2, or L_\infty norms in terms of visual similarity ?

When a classifier is built to discriminate images between 10 classes on an input space that contains much more that the 10 classes, is it surprising that some "small" perturbations can change the predicted classes ?
For most of input images the true class should be I don't know.

Overall I think that paper could be interesting to feed the discussions at the workshop.

---

### Official Review · AnonReviewer2 · 2018-03-09
**Feature squeezing is vulnerable, but this was already known**

**Rating:** 3
**Confidence:** 5

**Review:**

This paper shows that the "feature squeezing" defense against adversarial examples can be defeated using standard attacks, as long as the budget for the adversary is increased. Previous limitations on adversary strength may have been overly conservative, and there are visually indistinguishable adversarial examples that violate commonly-used constraints.

Analyzing the weaknesses of proposed defenses is a valuable contribution. I disagree with the comment thread about making the adversary stronger -- it's fine to evaluate against a stronger adversary (with looser constraints on the examples), as long as you can show that the resulting examples are sufficiently visually similar to the original images.

However, I thought that this paper fell short in two critical ways:

1. When the main innovation is a stronger or less-constrained adversary, it's very important to demonstrate that this adversary still generates good images. This was shown qualitatively with a small number of images, but since this claim is central, it needs a more thorough evaluation. This could be done with a user study, for example. (I was also confused by the epsilon=0.5 case -- shouldn't the adversary be able to create an all-gray image, which would always confuse a classifier? But I may have missed something, so I'll give it the benefit of the doubt.)

2. In the paper "Adversarial Example Defenses: Ensembles of Weak Defenses are not Strong" (He et al., 2017), the authors showed that feature squeezing is indeed vulnerable to adversarial attacks. This workshop paper needs to distinguish its contribution from previously published work. Feature squeezing was claimed to be effective, but that claim has already been disproven.

---

> ### Author Response · Authors · 2018-03-26
> **Response**
>
> Thank you for your comments. One clarification, when epsilon=0.5, for the L1/L2 optimized attacks (EAD, C&W), an all-gray adversarial example would not be generated as the L1/L2 distortion would be high. As we showed in earlier works, for an L1-optimized attacks, visually good adversarial examples can be constructed which have high epsilon because by minimizing L1, less pixels are modified.

---

### Public Comment · (anonymous) · 2018-02-27
**Increasing adversary strength**

Typically, phrasing like "stronger attack configurations" is used to talk about using strong (potentially adaptive) iterative attacks, but _within the threat model posed_.

Can "increasing adversary strength" as used in this paper be understood as stepping outside the threat model considered in the original Feature Squeezing paper, similar to what was done in this work (https://arxiv.org/abs/1710.10733)? If so, this paper does not invalidate any claims made in the Feature Squeezing paper, right? All this result shows is that Feature Squeezing doesn't work under the different threat model considered in this workshop paper.

---

> ### Author Response · Authors · 2018-02-27
> **Response**
>
> Thanks for your valuable comments. We use the same threat model as used in the original feature squeezing work (a powerful adversary with full access to the trained target model, but unaware of feature squeezing being performed on the operator's side). What we demonstrate is the hyperparameter setting used for the attacks in the original work is weak, and with stronger settings one can find adversarial examples which bypass the proposed joint detection method with minimal visual distortion. For example, $\kappa$ = 10 was used for the C&W attack in the original work, by increasing $\kappa$ we show that feature squeezing can be bypassed. Similarly for the $\epsilon$ parameter in PGD. The definition of adversary strength used in the paper is equivalent to the one used in the feature squeezing work.
>
> One note, regarding the paper cited, in its introductory work, no threat model was defined explicitly for the Madry Defense Model, it was solely tested in the white-box and black-box cases. In the competition, a black-box setting (transfer attack) was posed with an $L_\infty$ distortion constraint placed on the attacker. The cited paper argues that this constraint was used because the network as is could not be adversarially trained by PGD with larger $L_\infty$ distortion, and the addition of that constraint implies that visually imperceptible adversarial examples cannot be generated with $L_\infty$ distortion greater than the given constraint. The cited paper shows that by using $L_1$-based adversarial examples, despite their high $L_\infty$ distortion, visually imperceptible adversarial examples which successfully transfer to the Madry Defense Model can be generated.

---

> > ### Public Comment · (anonymous) · 2018-02-28
> > **Thanks**
> >
> > Thank you for the clarification. This is a pretty neat result!
> >
> > EDIT: I didn't read the response closely enough. I do see this as violating the threat model (/ an unbounded threat model is not reasonable or useful). Given the other discussion, this result doesn't seem particularly meaningful. Any unbounded attack should be able to achieve 100% success rate against a non-constant classifier.

---

> > > ### Author Response · Authors · 2018-03-01
> > > **Response**
> > >
> > > Thanks for your update. Per our first response, what we demonstrate is the hyperparameter setting used for the attacks in the original work is weak, and with stronger settings one can find adversarial examples which bypass the proposed joint detection method "with minimal visual distortion."  So, our generated adversarial examples bypass feature squeezing detection while also remaining visually similar to the original images. Please refer to Figures 1 and 2 for evidence.

---

> > ### Public Comment · (anonymous) · 2018-03-01
> > **Increasing $\epsilon$ changes the threat model**
> >
> > Increasing $\epsilon$ is stepping outside the threat model proposed by the original authors. You are not invalidating any claims by the authors of the feature squeezing paper.

---

> > > ### Author Response · Authors · 2018-03-01
> > > **Response**
> > >
> > > The threat model proposed by the original authors is the following: "In evaluating robustness, we assume a powerful adversary who has full access to a trained target model, but no ability to influence that model. For now, we assume the adversary is not aware of feature squeezing being performed on the operator’s side. The adversary tries to find inputs that are misclassified by the model using white-box
> > > attack techniques".  Therefore, there is indeed no adversary strength constraint (e.g.,  $\epsilon$ ) in the threat model.

---

> > > > ### Public Comment · (anonymous) · 2018-03-01
> > > > **No epsilon constraint**
> > > >
> > > > Here's a simple attack that also works in the "threat model" without an $\epsilon$ constraint:
> > > >
> > > > def compute_attack(image, target_class):
> > > >     return instance of target class
> > > >
> > > > No need to use anything as complicated as EAD.
> > > >
> > > > Unbounded distortion is not a reasonable or useful threat model.

---

> > > > > ### Author Response · Authors · 2018-03-01
> > > > > **Response**
> > > > >
> > > > > Per our first response, what we demonstrate is the hyperparameter setting used for the attacks in the original work is weak, and with stronger settings one can find adversarial examples which bypass the proposed joint detection method "with minimal visual distortion."  So, our generated adversarial examples bypass feature squeezing detection while also remaining visually similar to the original images. Please refer to Figures 1 and 2 for evidence.

---

> > > > > > ### Public Comment · (anonymous) · 2018-03-01
> > > > > > **Threat model**
> > > > > >
> > > > > > $\epsilon$ is part of the _threat model_. It's not a parameter that the attacker is able to choose.
> > > > > >
> > > > > > If the attacker is allowed to choose epsilon, why not choose epsilon=255 and use the provably optimal 1-line attack that I gave above that's guaranteed to defeat all current and future defenses?

---

> > > > > > > ### Author Response · Authors · 2018-03-01
> > > > > > > **Response**
> > > > > > >
> > > > > > > We would like to remind you that according to the feature squeezing paper, $\epsilon$ is not part of the threat model.
> > > > > > >
> > > > > > > In addition, if $\epsilon$=255 ($L_\infty$ constraint), as long as one can generate visually similar adversarial examples like those generated by the one-pixel attack (https://arxiv.org/pdf/1710.08864.pdf),  wouldn't one consider it as a powerful adversarial attack?

---

> > > > ### Public Comment · (anonymous) · 2018-03-01
> > > > **Response**
> > > >
> > > > Table 2 gives an evaluation of attacks under various $l_p$ and $\epsilon$ "distortion" levels, and thus gives a clear picture of the claims that the paper makes. Just because the authors of the paper do not label their threat model clearly does not mean that the threat model does not include an $\epsilon$ constraint, it is just implicit based on the evaluation. The authors make no claims on $\epsilon$ values larger than the values listed in Table 2.

---

> > > > > ### Author Response · Authors · 2018-03-01
> > > > > **Response**
> > > > >
> > > > > Based on your comment "Just because the authors of the paper do not label their threat model clearly does not mean that the threat model does not include an $\epsilon$ constraint", we suggest you contact the feature squeezing paper authors for justification. If they revise their threat model description with an explicit distortion constraint, we will follow up the changes accordingly.
> > > > >
> > > > > However, we would like to point out that even if the threat model only considers weak attacks (e.g., small $\epsilon$ constraint), our Figures 1 and 2 still clearly demonstrate the existence of visually similar adversarial examples when one makes the attack stronger, by increasing the adversary strength. We believe the fundamental question is: can feature squeezing be effective against stronger attacks when the resulting adversarial examples are still similar to the original ones?

---

### Decision · Program_Chairs · 2018-03-20
**ICLR 2018 Workshop Acceptance Decision**

**Decision:**

Reject

**Comment:**

Based on the reviews, this paper has not been accepted for presentation at the ICLR workshop. However, the conversation and updates can continue to appear here on OpenReview.